# Diffusion Tensor Imaging Changes Do Not Affect Long-Term Neurodevelopment following Early Erythropoietin among Extremely Preterm Infants in the Preterm Erythropoietin Neuroprotection Trial

**DOI:** 10.3390/brainsci11101360

**Published:** 2021-10-16

**Authors:** Janessa B. Law, Bryan A. Comstock, Todd L. Richards, Christopher M. Traudt, Thomas R. Wood, Dennis E. Mayock, Patrick J. Heagerty, Sandra E. Juul

**Affiliations:** 1Division of Neonatology, Department of Pediatrics, University of Washington, Seattle, WA 98195, USA; traudtchris@msn.com (C.M.T.); tommyrw@uw.edu (T.R.W.); mayock@uw.edu (D.E.M.); sjuul@uw.edu (S.E.J.); 2Department of Biostatistics, University of Washington, Seattle, WA 98195, USA; bac4@uw.edu (B.A.C.); heagerty@uw.edu (P.J.H.); 3Department of Radiology, University of Washington, Seattle, WA 98195, USA; toddr@uw.edu

**Keywords:** diffusion tensor imaging, preterm, erythropoietin, clustering coefficient

## Abstract

We aimed to evaluate diffusion tensor imaging (DTI) in infants born extremely preterm, to determine the effect of erythropoietin (Epo) on DTI, and to correlate DTI with neurodevelopmental outcomes at 2 years of age for infants in the Preterm Erythropoietin Neuroprotection (PENUT) Trial. Infants who underwent MRI with DTI at 36 weeks postmenstrual age were included. Neurodevelopmental outcomes were evaluated by Bayley Scales of Infant and Toddler Development (BSID-III). Generalized linear models were used to assess the association between DTI parameters and treatment group, and then with neurodevelopmental outcomes. A total of 101 placebo- and 93 Epo-treated infants underwent MRI. DTI white matter mean diffusivity (MD) was lower in placebo- compared to Epo-treated infants in the cingulate and occipital regions, and occipital white matter fractional isotropy (FA) was lower in infants born at 24–25 weeks vs. 26–27 weeks. These values were not associated with lower BSID-III scores. Certain decreases in clustering coefficients tended to have lower BSID-III scores. Consistent with the PENUT Trial findings, there was no effect on long-term neurodevelopment in Epo-treated infants even in the presence of microstructural changes identified by DTI.

## 1. Introduction

Advances in neonatology have led to unprecedented improvements in neonatal survival such that infants born at 22-0/7 to 25-6/7 weeks’ gestation now have a 70% survival rate [1]. Unfortunately, neurodevelopmental outcomes for extremely preterm (EP) infants have not improved at the same rate. While a recent report suggests that an increasing percentage of those born preterm have no major disabilities, up to 40% of survivors born at less than 28 weeks of gestation still develop one or more complications including cerebral palsy, intellectual disability, visual or auditory deficits [2,3,4,5]. New neuroprotective strategies are therefore critically needed to further improve outcomes in these vulnerable children.

At least 50% of infants born prematurely have qualitative evidence of cerebral white matter injury on magnetic resonance imaging (MRI) at term-equivalent age [6,7]. These gross white matter findings are correlated with increased risk of neurodevelopmental impairment (NDI), but they are limited in their ability to describe the exact structural and functional alterations responsible for these outcomes [8,9,10,11,12]. Diffusion tensor imaging (DTI) is an MRI technique that aims to clarify the microstructure of the brain by measuring the directionality (fractional anisotropy, FA, and mode, MO) and rate (mean diffusivity, MD) of water diffusion within the structures of an MRI voxel [13]. Generally, in white matter, FA increases with increased myelination and MD decreases with maturation and increasing structural integrity [14,15]. In contrast, FA in grey matter decreases during the last trimester as does MD, reflecting an increase in cellular and synaptic complexity and density [16].

Microstructural descriptions provide a valuable insight into the anatomic effect of injury on premature brain development; however, it is not well understood how these changes affect subsequent brain function. To address this question, DTI imaging can be used to model the brain as a complex grid of interconnected regions (nodes) with visual and quantitative characterization of the organization between nodes, referred to as connectivity [17]. One component of connectivity is the ability of a node to transfer information within a network of nodes, which is quantified by clustering coefficients, with a high clustering value indicative of greater network transfer efficiency [18]. This approach to analysis provides a vital framework to elucidate the relationship between brain structure and networking [19,20]. Emerging data link connectivity to neurodevelopment and behavior in children and adults born preterm, though studies establishing connectivity reference values in extremely preterm well infants as well as studies assessing the association between connectivity and neurological outcomes in this high-risk cohort are still needed [21,22,23].

The PENUT Trial was a randomized placebo-controlled trial studying the effects of erythropoietin (Epo) on death or severe NDI in infants born extremely preterm [24]. Using a subset of PENUT participants who underwent MRI with DTI, our first objective was determine DTI values in this patient population and to evaluate the effect of Epo on those parameters when measured at 36 weeks postmenstrual age (PMA). Second, we aimed to explore the relationship between these values and neurodevelopmental outcomes at 2 years corrected age (CA) to help identify those infants at highest risk for NDI. We hypothesized that Epo would result in DTI values consistent with increased axonal maturation, myelination, and connectivity, and that these DTI findings would correspond to improved performance on neurodevelopmental assessments at 2 years CA.

## 2. Materials and Methods

### 2.1. Eligibility and Enrollment

The PENUT Trial included 19 sites across the United States, and enrolled inborn infants 24-0/7 to 27-6/7 weeks’ gestation between December 2013 and September 2016. Patients were excluded based on known life-threatening disorders, chromosomal anomalies, disseminated intravascular coagulopathy, twin-twin transfusion, hematocrit > 65%, hydrops fetalis, or known congenital infection [24]. Infants randomized to treatment received Epo intravenously at a dose of 1000 U per kilogram of body weight every 48 h for a total of six doses, followed by a maintenance dose of 400 U per kilogram three times per week by subcutaneous injection through 32 completed weeks PMA. Participating infants at a designated subset of PENUT Trial recruitment sites who underwent MRI with DTI at 36–37 weeks PMA and were evaluated for neurodevelopmental outcomes at 22–26 months corrected age were included in this analysis. Demographic and clinical characteristics of enrolled infants surviving to 36–37 weeks PMA, but did not undergo MRI, are also presented.

### 2.2. Ethics

The PENUT Trial enrolled infants after informed parental consent. This study was IRB approved at all participating sites, and registered with ClinicalTrials.gov (NCT01378273) and the FDA (IND#12656) [24].

### 2.3. Imaging Protocol

Five study sites acquired data from Siemens 3T MR scanners and three acquired data from Philips 3T MR scanners. Acquisition protocols were similar, although not identical, between scanners, and the number of diffusion gradient angles obtained was limited by the duration of scan time tolerated by the primary study protocol. Phantoms were scanned using the same infant MRI protocol to test for scanner quality at all sites. At the beginning of the trial, an MRI scan of the same person at all sites (SEJ) was performed to improve comparison across centers. Whole brain DTI FA values were compared taking an average and a standard deviation and then calculating a percent of the standard deviation of the average across all sites (average = 0.198, standard deviation = 0.0126). The FA means of the Siemens and Philips scanners were 0.198 and 0.194, respectively, with a two-tailed *p* value of 0.32 (no significant difference).

Siemens MR acquisition protocol: The diffusion-weighted MR sequence was collected in the axial plane with an echo-planar diffusion weighted spin-echo pulse sequence with the following parameters: echo spacing 0.71 ms, EPI factor 112, TR/TE 7000/73 ms, voxel size 2 × 2 × 2 mm^3^, 55 slices, interleaved acquisition of slices, field of view 224 mm, fat suppression on, 30 different diffusion gradient angles + one non-diffusion volume, b-value 1000 s/mm^2^, 2 averages.

Philips MR acquisition protocol: The diffusion-weighted MR sequence was collected in the axial plane with an echo-planar diffusion weighted spin-echo pulse sequence with the following parameters: TR/TE 7733/73 ms, echo-train-length 45, voxel size 1.4 × 1.4 × 2 mm^3^, 55 slices, field of view 180 mm, 32 different diffusion gradient angles + one non-diffusion volume, b-value 1000 s/mm^2^, reconstructed matrix 128 × 128 × 55.

### 2.4. DTI Data Processing

As described in the FMRIB Software Library v6.0, motion and distortion of diffusion-weight images were corrected for using FSL eddy software [25,26]. For model DTI tensor fit, FSL’s dtifit was used and the resulting tensor was median filtered using the fslmaths -fmedian option. The resulting output files were FA, MO, MD, L1, L2, and L3 maps. The directionality of water movement is provided by FA, the degree to which the diffusion is isotropic (value of 0) or anisotropic (higher value up to 1), and MO, the measure of tensor shape as either planar (value of −1) or linear (up to a value of 0). MO complements FA by providing insight into the extent of crossing fibers and has been increasingly linked to long-term behavioral outcomes [27,28].

MD is a mean measure of the magnitude of water diffusion detected across three gradient directions (eigenvalues L1, L2, and L3) in mm2/s. Diffusivity along and parallel to the principal axis is measured by L1 (axial diffusivity, AD) and has been associated with the axon diameter, whereas diffusivity in directions perpendicular to the principal axis of diffusion are measured by the average of L2 and L3 (radial diffusivity, RD) and has been associated with the degree of myelination and number of branching exiting fibers [29,30,31]. Exact commands for eddy and dtifit can be found in the Appendix B.

Following transformation of all subjects into the same space for direct comparison, voxelwise statistical analysis of the DTI data was carried out using a two-group design with statistical significance defined as *p* < 0.05 [32,33]. Co-registration of the FA maps from all subjects was performed using software BuildTemplate from Advanced Normalization Tools (ANTs, http://stnava.github.io/ANTs/, accessed on 10 February 2019). Instead of relying on standard tract-based spatial statistic (TBSS) skeleton projection (which is dependent on adult white matter/DTI characteristics), this procedure builds a template from all of the subjects and also co-registers the individual FA maps to the same whole brain template. The ANTs procedure has been validated as a rigorous non-linear approach to co-register subject brains to a subject template [34,35]. Indeed, this method has been shown to outperform TBSS skeleton analysis with improved sensitivity and specificity when detecting group differences [36]. ANTS software MeasureImageSimilarity metric was used to determine goodness of co-registration where 1.0 is considered a perfect score: Mean metric for 219 subjects = 0.968, standard deviation for metric = 0.0024. The co-registered FA maps were combined into a 4-dimensional volume which was fed into software FSL Randomise to extract the ROI values and rigorously test for group differences between Epo and placebo-treated infants [37,38]. 1000 permutations were used. These permutations are also well explained by the FMRIB Software Library v6.0 and used routinely for thresholding on statistic maps. FSL Randomise produces a *p*-value map corrected for multiple-voxel comparisons using the threshold-free cluster enhancement (TFCE) option. Per Spisák et al., “TFCE integrates cluster information into voxel-wise statistical inference to enhance detectability of neuroimaging signals” [39].

Region of interest (ROI) analysis and identification of brain region anatomy of the infant MRI brains was based on atlases and templates created in the laboratory of Dr. John E. Richards [40]. While the infants used to create the atlas were approximately 4 weeks of age older than our cohort, the advantage of using these templates, relative to study-specific templates, is that they improve brain region specificity and provide anatomical brain region information.

Two ROIs in the white matter (cingulate white matter near the cingulate gyrus and occipital white matter) and three in the grey matter (bilateral basal ganglia and occipital regions) as described in the Automated Anatomical Labelling atlas were evaluated [41]. The ROIs in the occipital and cingulate areas were based on significant large clusters (with corrected *p* values < 0.02) found on the axial diffusivity TFCE randomise statistical map created by comparing the 2 main groups. These ROIs/clusters were near AAL regions described as follows: AAL region 50 superior occipital gyrus, AAL region 52 middle occipital gyrus, AAL region 34 middle cingulate cortex, AAL region 20 supplemental motor area. The ROIs chosen in the basal ganglia were based on prior reports of brain injury found in the basal ganglia, and these regions were close to AAL region 73 Putamen Left and AAL region 74 Putamen Right [3]. The white matter near the cingulate gyrus projects inputs from the neocortex and thalamus to the entorhinal cortex and plays an important role in learning, memory, and emotional development. While assisting a broad variety of functions, the basal ganglia grey matter is primarily responsible for integration of cortical signals into voluntary motor movements, cognition and decision-making, and emotion. The occipital lobe white and grey matter regions are critical for vision and image processing. Appendix A shows the location of the ROIs.

As an indicator of network segregation, clustering coefficient measures functionally distinct networks that have been linked to separate and measurable cognitive processes [42,43]. We chose to focus specifically on this parameter as changes to these cognitive processes have been demonstrated in the pediatric literature, with strong links to measurable neurodevelopmental and behavioral outcomes including internalizing and externalizing behaviors as measured by the Child Behavior Checklist at 2 and 4 years of age, as well as reading dysfunction in school-age children [44,45]. Preliminary investigations evaluating the structural connectome of healthy or mildly preterm infants have begun to shed light on the natural development of these structural networks [46,47]. The effect of extremely preterm delivery on this process is unknown, and normative values in this patient population at term-equivalent age are missing, thus determining reference values in for this population is needed. Clustering coefficients were measured using FSL’s probtrackx2 software with network option enabled and using seed points derived from the JHU MNI SS WMPM atlas, which was also adapted for the infant brain [48]. Exact commands used can be found in the Appendix B. Matlab software “Brain Connectivity Toolbox” (https://sites.google.com/site/bctnet/construction, accessed on 15 February 2019) uses clustering coefficients to perform the complex network/graph theory analysis, as described in Rubinov and Sporns [49]. DTI quality control was performed using DTIprep which is software that checks for artifacts caused by eddy-currents, head motion, bed vibration and pulsation, venetian blind artifacts, as well as slice-wise and gradient-wise intensity inconsistencies [50]. The clustering coefficients of the regions were based on the full-brain connectivity network regions thresholded at 10% sparsity—network efficiency studies support thresholds from 10% to 50%, and a 10% threshold has been previously utilized for graph analyses in children [45,51]. Brain network figures were generated using software BrainNet Viewer as described by Xia et al. [52].

### 2.5. Neurodevelopmental Assessments

Neurodevelopmental outcomes at 22–26 months CA were evaluated using the Bayley Scales of Infant and Toddler Development, 3rd Edition (BSID-III). Individuals performing the BSID-III assessments were centrally certified and blinded to the child’s medical history, treatment arm, and brain imaging studies.

### 2.6. Statistical Analysis

A modified intent-to-treat (ITT) approach was used in all analyses, with all randomized infants who received the first dose of study treatment included in the analyses. For all statistical comparisons between groups, Generalized Estimating Equations (GEE) with robust standard errors and a working independence correlation structure were used to account for inclusion of infants from a multiple gestation [53]. Complications and comorbidities between birth and 36 weeks’ PMA, and outcomes at age 2 were examined. Given the association between iron dosing and neurodevelopmental outcomes as previously published, we chose to evaluate ferritin levels between groups as a surrogate of iron sufficiency [54].

The primary analysis was a comparison of DTI values (extracted by the FSL software) between randomized treatment groups. Specifically, a GEE-based Wald test was used to examine differences in diffusion values and clustering coefficients between treatment groups, with adjustment for gestational age (GA) at birth used to stratify treatment randomization (24 + 0 to 25 + 6 vs. 26 + 0 to 27 + 6 in weeks + days of GA), sex, and scanner type used at the enrolling hospital. In secondary analyses, for any DTI measurements found to significantly differ in the main comparison, an interaction term between treatment and GA at birth was used to explore treatment effect moderation by GA.

Adjustment for multiple testing among the 15 DTI measurements (FA, MD, and MO for each of the five regions analyzed) was handled with the Bonferroni-Holm procedure, with the aim of controlling the overall type I error rate at 0.05. L1, L2, and L3 values were excluded from analysis as the correlation coefficient between these gradients and MD in all ROIs was close to 1 (data not shown). *p*-values generated from the m = 15 tests of DTI diffusion values by treatment group were sorted from smallest to largest and compared against nominal levels of 0.05/m, 0.05/(m−1), 0.05/1, where statistical significance was declared for all *p*-values smaller than the smallest indexed *p*-value such that Pk > 0.05/(m + 1−k). Clustering coefficients (*N* = 66) were assessed overall and by region using a Manhattan plot of the *p*-values, with a similar Bonferroni-Holm correction procedure for declaration of statistical significance. Tests of treatment interactions with GA were considered exploratory analyses.

GEE analyses were used to estimate the association between DTI measures and BSID-III cognitive outcomes at 2 years CA with adjustment for treatment assignment, GA, sex, and study recruitment site. All statistical analyses were performed using the R statistical software package (version 3.3.0, Vienna, Austria).

### 2.7. Role of the Funding Source

Funders did not have any role in study design, data collection, data analyses, interpretation, or writing of report.

## 3. Results

### 3.1. Enrollment and Group Demographics

Of the 741 infants enrolled in the PENUT Trial, 469 infants were enrolled across eight designated MRI sites (229 placebo-treated and 240 Epo-treated), with approximately equal stratification within each group for 24–25 week and 26–27 week GAs. One hundred and one infants in the placebo group and 93 infants in the Epo group underwent an MRI (132 Siemens, 62 Phillips) and were included. Of these, 81 and 73 underwent 2 year neurodevelopmental assessments, respectively (Figure 1).

As entry into the MRI cohort was conditional upon surviving until 36 weeks PMA, demographic and clinical characteristics were compared between treatment groups (Epo vs. Placebo) within the imaging cohort as well as between the MRI cohort and infants who were enrolled at MRI recruitment sites, survived to 36 weeks PMA, but were not in the MRI cohort. Table 1 shows these comparisons.

While there were no substantial differences between the treatment groups within the MRI cohort, those in the MRI group more commonly had a Hispanic mother (34% vs. 22%; *p* < 0.05) and were more likely to demonstrate attributes associated with survival including delayed cord clamping (60% vs. 37%; *p* < 0.01), higher mean birth weight (831 g vs. 783 g; *p* < 0.01), and higher Apgar score at 5 min (6.6 vs. 6.0; *p* < 0.01).

### 3.2. Comparison of Adverse Events across Treatment Groups

Given that both inflammation and maturity can affect DTI values, we queried whether the two treatment groups were similar in the postnatal complications of prematurity they experienced. Table 2 shows the incidence of common inflammatory complications of prematurity for the MRI cohort and the non-MRI cohort.

There were no statistically significant differences between the Epo and placebo groups or between the MRI and non-MRI cohorts in necrotizing enterocolitis (NEC), spontaneous intestinal perforation (SIP), or retinopathy of prematurity (ROP). When compared to the non-MRI cohort, the MRI cohort had significantly fewer infants with culture proven sepsis (3.1% vs. 12%; *p* = 0.003) or grade III/IV intraventricular hemorrhage (IVH) (3.1% vs. 16%; *p* < 0.001).

Iron deficiency evaluated by serum ferritin was also queried as significant iron deficiency can result in delayed myelination [55,56]. In contrast to the inflammatory insults above, moderate (<76 µg/mL) and severely low (<40 µg/mL) ferritin levels were present significantly more often in infants treated with Epo compared to placebo (Table 2). Chronic lung disease (CLD) did not differ between the Epo and placebo groups or between the MRI and non-MRI cohorts. BSID-III cognitive (95.4 vs. 87.4; adjusted difference (95% CI): −6.2 (−9.7 to −2.7); *p* < 0.001) and motor (93.8 vs. 85.7; adjusted difference (95% CI): −6.6 (−10.1 to −3.1); *p* < 0.001) scores at 2 years of age were significantly higher in the MRI cohort compared to the non-MRI cohort, again reflecting a healthier cohort overall. There was no statistically significant difference in BSID-III language scores between groups (adjusted difference (95%CI): −0.6 (−6.1 to −4.9); *p* = 0.73).

### 3.3. Comparison of DTI Measures

#### 3.3.1. DTI Measures by Treatment Group

Figure 2 shows an unadjusted DTI ANTs analysis comparing diffusion values (in this case MD) between treatment groups prior to additional statistical analysis (including corrections for sex and scanner type) described in the Methods section above.

The Manhattan plot used to detect areas of statistically significant differences in diffusion values is shown in Figure 3a. White matter MD was lower in placebo-treated infants compared to those treated with Epo in both cingulate (in mm^2^/s, mean ×10,000: 13.25 vs. 13.70; adjusted difference: 0.49; 95% CI: 0.24 to 0.73; *p* < 0.0001) and occipital (in mm^2^/s, mean ×10,000: 14.84 vs. 15.39; adjusted difference: 0.54; 95% CI: 0.27 to 0.80; *p* < 0.0001) ROIs (Figure 3c,d). No other diffusion measures in the white or grey matter ROIs were different between treatment groups, and there was no correlation between Epo level and any of the MD values among infants treated with Epo (data not shown). FA and MO measures did not differ between groups.

There were no statistically significant differences in clustering coefficients between placebo- and Epo-treated infants (Figure 3b). An example of a connectivity map from which clustering coefficients are derived is provided in Appendix A.

#### 3.3.2. DTI Measures by Gestational Age

The Manhattan plot used to detect areas of statistically significant differences in diffusion values by GA is shown in Figure 4a. No diffusion measures in the white or grey matter ROIs were statistically different by GA after adjustment for multiple testing. However, occipital white matter FA was lower in the 24–25 week infants (mean ×10,000: 782.02) compared to 26–27 week infants (mean ×10,000: 821.69; adjusted difference: 35.89; 95% CI: 12.44 to 59.35; *p* = 0.0027) despite obtaining the MRIs at the same PMA (Figure 4c).

The left thalamus had higher clustering coefficients in infants born at 24–25 weeks’ GA (mean ×100: 17.10) relative to those born at 26–27 weeks’ GA (mean ×100: 15.66; adjusted difference: −1.35; 95% CI: −2.20 to −0.49; *p* = 0.0019) (Figure 4b,d), though this difference did not reach the multiple testing threshold for statistical significance.

#### 3.3.3. DTI Measures by GA and Treatment Group

There were no significant treatment by GA interactions in any white or grey matter ROI DTI diffusion values (Figure 5a).

Treatment-related differences in clustering coefficients were significantly moderated by GA (Figure 5b). The right precentral cortex in Epo-exposed infants born at 24–25 weeks had significantly lower clustering coefficients (interaction: 4.30; 95% CI: 2.38 to 6.22; *p* < 0.0001) (Figure 5c).

#### 3.3.4. Association between DTI Measures and 2 Year Neurodevelopment

Neither white nor grey matter DTI diffusion values were associated with BSID-III cognitive, motor, or language composite scores at age 2 (Figure 6a, Figure 7a, Figure 8a).

Increasing clustering coefficients were positively associated with BSID-III motor scores in the left middle occipital lobe (occipital mid left; coefficient: 1.4; 95% CI: 0.4 to 2.4; *p* = 0.005) (Figure 6b,c) and in the right paracentral lobule area (coefficient: 1.2; 95% CI: 0.3 to 2.1; *p* = 0.009) (Figure 6b,d). Increasing clustering coefficients in the right medial superior frontal gyrus (right medial superior frontal gyrus; coefficient: 1.5; 95% CI: 0.5 to 2.4; *p* = 0.002) and right paracentral lobule area (coefficient: 1.4; 95% CI 0.4 to 2.5; *p* = 0.007) were positively associated with cognitive scores on the BSID-III (Figure 7b–d). Each 0.01-point difference in clustering coefficient was associated with a 1.5-point increase (95% CI: 0.5 to 2.4) in cognitive score (Figure 7c). BSID-III language scores were positively associated with right medial superior frontal gyrus (coefficient: 1.4; 95% CI: 0.5 to 2.2; *p* = 0.002) (Figure 8b,c) and in the right superior occipital lobe (coefficient: 0.8; 95% CI: 0.2 to 1.4; *p* = 0.008) (Figure 8b,d). None of the associations between BSID-III scores and clustering coefficients met the multiple testing threshold for statistical significance.

## 4. Discussion

Epo is an important trophic factor during fetal brain development and has robust neuroprotective effects in preclinical models of brain injury [57,58,59,60,61,62,63]. Neuroprotective effects in preclinical models include decreasing inflammation, excitotoxicity, and oxidative injury while promoting erythropoiesis, neurogenesis and oligodendrogenesis [61,64,65,66,67]. Initial data on the neuroprotective effects of Epo were obtained in rodent models of brain injury, which were later supported by data derived from fetal sheep, piglet, and nonhuman primate models of neonatal brain injury [68,69]. It is important to note that most if not all models of neonatal brain injury are acute, and do not accurately model the prolonged postnatal period during which preterm infants are exposed to exogenous and endogenous stimuli including hypoxia, hypoxia-ischemia, hyperoxia, inflammation, excitotoxicity, and an excess of free-radicals. Nutritional deficiencies such as iron deficiency, as well as exposure to pain, light, noise, drugs, and other factors in the neonatal intensive care environment also play a role in modifying extra-uterine development [70].

Our hypothesis, that Epo treatment would improve MRI indicators of myelination and connectivity in extremely preterm infants, was not supported. We anticipated an increase in white matter FA, and decreased MD, both of which are associated with increased myelination. Indeed, we observed the opposite, with no difference in FA, and Epo-treated infants showing increased white matter MD. This is in contrast to findings from the Swiss EPO Neuroprotection Trial in which 165 infants (77 Epo, 88 placebo) with mean GA at birth of 29 weeks underwent MRI at term equivalent age and showed fewer areas of gross white matter injury and increased FA in the group treated with Epo [70,71]. That cohort was more mature and received a higher dose of Epo (3000 U/kg/dose) over a shorter duration (3 doses within the first 48 h of life) than infants in the PENUT Trial. Similarly, Yang et al. found higher FA values in a cohort of 81 infants (42 Epo, 39 placebo) born ≤ 32 weeks’ GA; however, their cohort was also older, making direct comparison challenging [72]. Our findings were more consistent with the BRITE study which enrolled similar infants and showed no significant effect on FA in infants treated with erythropoietin stimulating agents; however, imaging occurred much later in that cohort (3.5–4 years of age) [73]. Exposure to medications such as postnatal steroids, opiates, and benzodiazepines are detrimental to neurodevelopment and may have masked the neuroprotective effect of Epo in our cohort [74].

We speculate that iron deficiency, and not Epo itself, may have contributed to the increased white matter MD values seen in our extremely preterm Epo-treated patients. Iron is required for normal brain maturation as it is essential for synaptogenesis, myelination, and dopamine synthesis, and iron deficiency during critical windows of development (fetal life through infancy) may lead to irreversible developmental deficits [56,75,76]. We also found a positive association between cognitive outcomes and iron dose at 2 months of age [54]. Oligodendrocytes are particularly vulnerable to iron deficiency during development as they rely on iron-requiring enzymes for early differentiation [77,78]. We previously reported that despite receiving more enteral and IV iron supplementation, more Epo- than placebo-treated infants had evidence of moderate or severe iron deficiency during their hospitalization, potentially contributing to oligodendrocyte injury and white matter structural changes as seen by DTI [4,79]. This is likely due to increased iron utilization associated with increased erythropoiesis in Epo-treated infants as manifested by fewer mean blood transfusions (3.5 vs. 5.2) and higher mean hematocrits (36.9% vs. 30.4%) in the Epo- vs. placebo-treated infants [79]. Investigation into outcomes of iron sufficient versus iron deficient infants in this cohort is ongoing.

Injury to cerebral white matter from cytotoxic edema and ischemia have also been shown to cause demyelination and oligodendrocyte death during early development [80]. These changes are pronounced in infants who suffer significant inflammatory insults such as chorioamnionitis or NEC [81]. While there was no statistical difference in complication incidence or 2 year neurodevelopmental outcomes between placebo- and Epo-treated infants, the MRI cohort as a whole suffered fewer acute inflammatory insults than the non-MRI cohort, indicating a possible selection bias to undergo MRI.

Aside from acute inflammatory complications, extremely premature infants also remain particularly vulnerable to sustained inflammatory states such as those caused by recurrent hypoxic insults, oxidative stress, hypotension, and CLD [15]. Several studies have demonstrated delayed axonal maturation and myelination in vulnerable areas of the brain even in relatively healthy infants born very- and extremely-preterm compared to term controls [81,82,83,84]. These delays in axonal maturation and myelination of cerebral white matter have been linked to lower scores on motor and behavioral assessments [85,86]. Although we also demonstrated that infants born at 24–25 weeks’ gestation have delays in measures of brain development (FA and MD) compared to infants 26–27 weeks’ gestation at birth, these structural changes were not linked to any significant changes in BSID-III scores at 2 years of age. It is possible that as BSID-III scores may overestimate neurodevelopmental scores, thus our analysis may have missed an association between DTI measures and infants with some level of NDI not identified by BSID-III testing [87,88,89].

We found it notable, however, that infants with decreased clustering coefficients in specific brain regions tended to have worse neurodevelopmental outcomes. In our study, BSID-III motor scores positively associated with increased clustering coefficients in the left middle occipital lobe and in the right paracentral lobule area. While these regions serve multiple functions, they each play a significant role in motor function [90,91]. Similarly, BSID-III cognitive scores positively associated with higher clustering coefficients in the right medial superior frontal gyrus and the right paracentral lobule area, both of which are involved in cognitive control of motor function (motor planning based on environmental context in the medial superior frontal gyrus and executive motor inhibition in the paracentral lobule) [92,93]. Lastly, BSID-III language scores positively associated with higher clustering coefficients in the right medial superior frontal gyrus and in the right superior occipital lobe. These findings are consistent with neuroanatomy demonstrating that the superior medial superior frontal gyrus contains a connection between the superior frontal language area and Broca’s regions of the brain, and there is emerging data to support the language processing potential of the visual association area within the occipital cortex [94,95]. While these trends did not meet statistical significance after multiple corrections, these data may indicate that the type of neurodevelopmental impairment detected by BSID-III at 2 years of age for infants born extremely preterm may be in part determined by the structural cerebral connectivity pattern identified early in neonatal life.

There were several limitations to this study. First, scanner gradient and scanner RF imperfections cause DTI measurements to be imperfect, resulting in greater signal noise in the more peripheral, cortical regions of the brain [96]. However, MRI-based parameters derived from these areas were not associated with any of the assessed outcomes in this study. Second, we were unable to correct for crossing fibers as this requires a greatly extended scan time (allowing for 64 diffusion directions), which is often not tolerated in neonates and was not the primary aim of the study imaging protocol. Inter-scanner comparisons were limited by differences in voxel size, although scanner type was adjusted for in this analysis. Additionally, we acknowledge that rapid brain growth and maturation during the neonatal period make the ROI templates used in this study from infants scanned approximately 4 weeks later than our cohort (40 weeks vs. 36 weeks PMA) an imperfect comparison that may have contributed to our findings. Finally, while the first dose of Epo was given within the first 24 h of life, the final dose was not received until 32 weeks 6 days PMA. It is possible that the white matter effects would not have been visible by the time of the 36 weeks PMA MRI. Similarly, FA values continue to increase as infants approach full term, so perhaps an MRI performed at a later age would be better able to detect differences between treatment groups [97].

## 5. Conclusions

In summary, extremely preterm infants remain at high risk for neurodevelopmental impairment. Early treatment with Epo did not provide structural protection of cerebral white matter as assessed by DTI. While commonly used DTI measures of white matter integrity (FA, MO, and MD) were not linked to neurodevelopmental outcomes, changes in cerebral clustering coefficients at 36 weeks’ PMA were positively associated with BSID-III motor, cognitive, and language scores at 2 years of age, a link which warrants further study. Investigation into neuroprotective therapies is ongoing; advanced DTI techniques may provide insight into connectivity-directed therapies to optimize neurodevelopmental outcomes in infants born extremely preterm.

## Figures and Tables

**Figure 1 brainsci-11-01360-f001:**
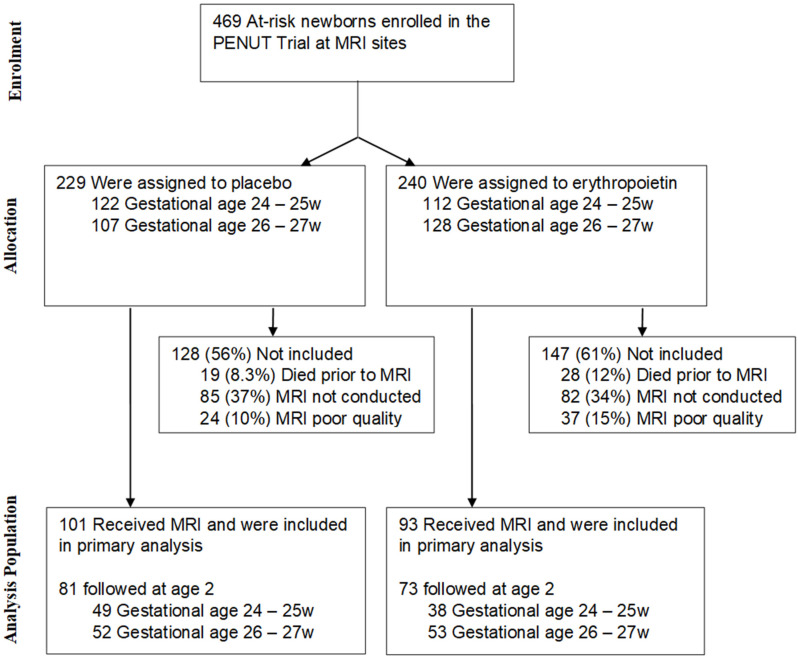
CONSORT diagram of PENUT MRI cohort.

**Figure 2 brainsci-11-01360-f002:**
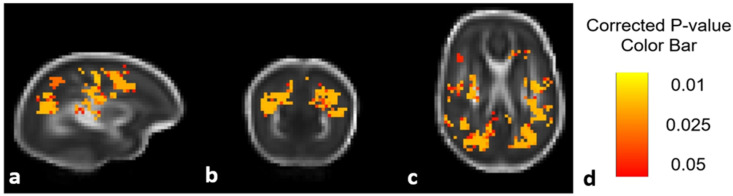
DTI comparison of Epo-treated and placebo-treated groups using ANTs analysis. MD differences seen in the sagittal (**a**), coronal (**b**), and axial (**c**) views. Highlighted areas signal regions in which the Epo treated group had significantly higher MD values compared to the placebo group prior to evaluation with multiple corrections. *p*-value color bar shown in (**d**).

**Figure 3 brainsci-11-01360-f003:**
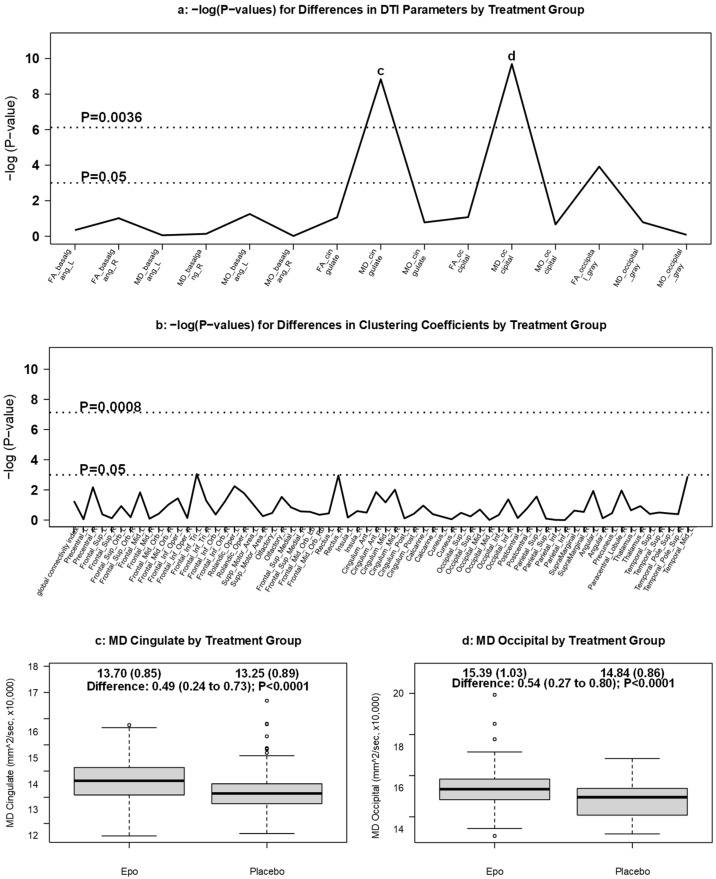
DTI measures by treatment group. Manhattan plot of [GEE-based Wald test] -log (*p*-values) for differences in (**a**) regional DTI diffusion values and (**b**) clustering coefficients by treatment group. Panels (**c**,**d**) display boxplots of ROIs with MD values found to be significantly different in the cingulate and occipital white matter ROIs, respectively, by treatment group.

**Figure 4 brainsci-11-01360-f004:**
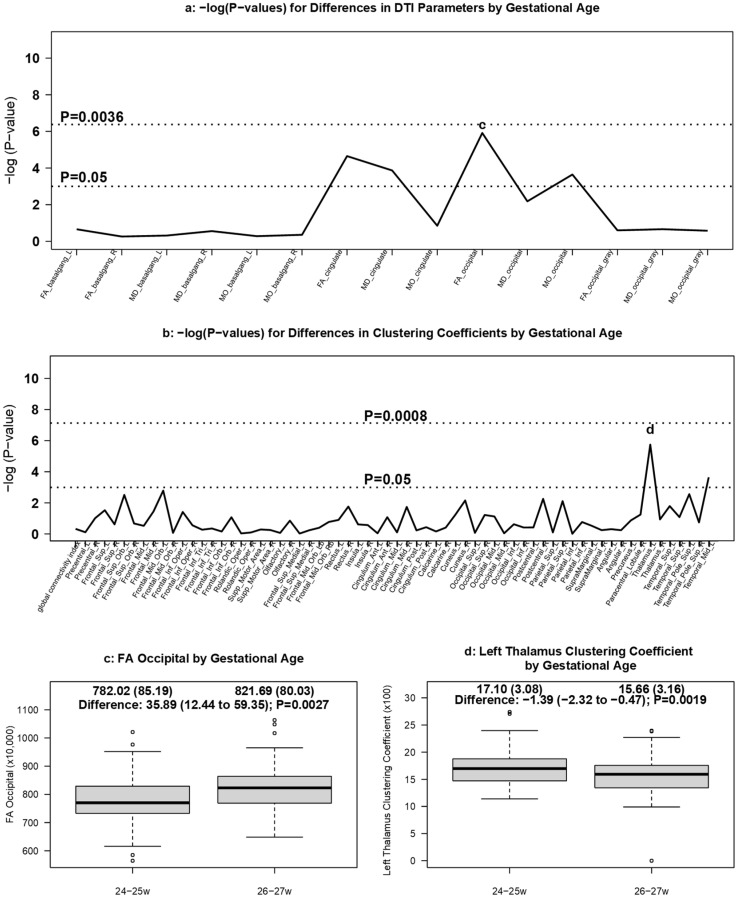
DTI measures by gestational age. Manhattan plots of [GEE-based Wald test] -log (*p*-values) for differences in (**a**) regional DTI diffusion values and (**b**) clustering coefficients by GA. Panel (**c**) displays boxplots of differences in the FA measurements in the occipital white matter region by GA, and panel (**d**) presents boxplots of clustering coefficients in the left thalamus found to be different by GA.

**Figure 5 brainsci-11-01360-f005:**
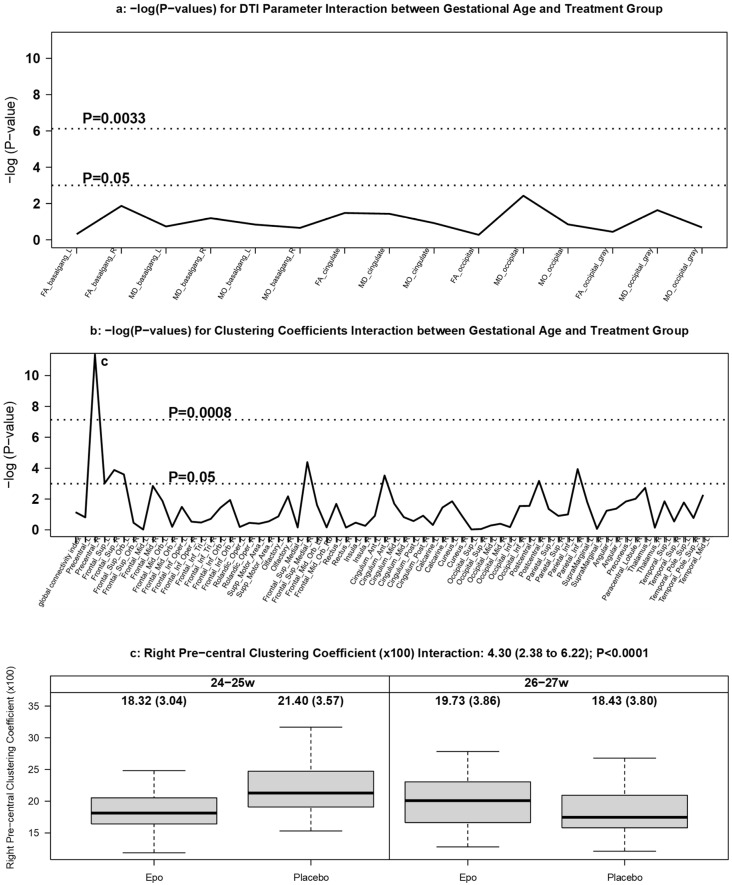
DTI measures by gestational age and treatment group. Manhattan plots of [GEE-based Wald test] -log (*p*-values) for differences in (**a**) regional DTI diffusion values and (**b**) clustering coefficients for the interaction between GA and treatment group. Panel (**c**) displays boxplots of clustering coefficients in the right precentral region that were significantly different [GEE-based Wald test] by GA and treatment group.

**Figure 6 brainsci-11-01360-f006:**
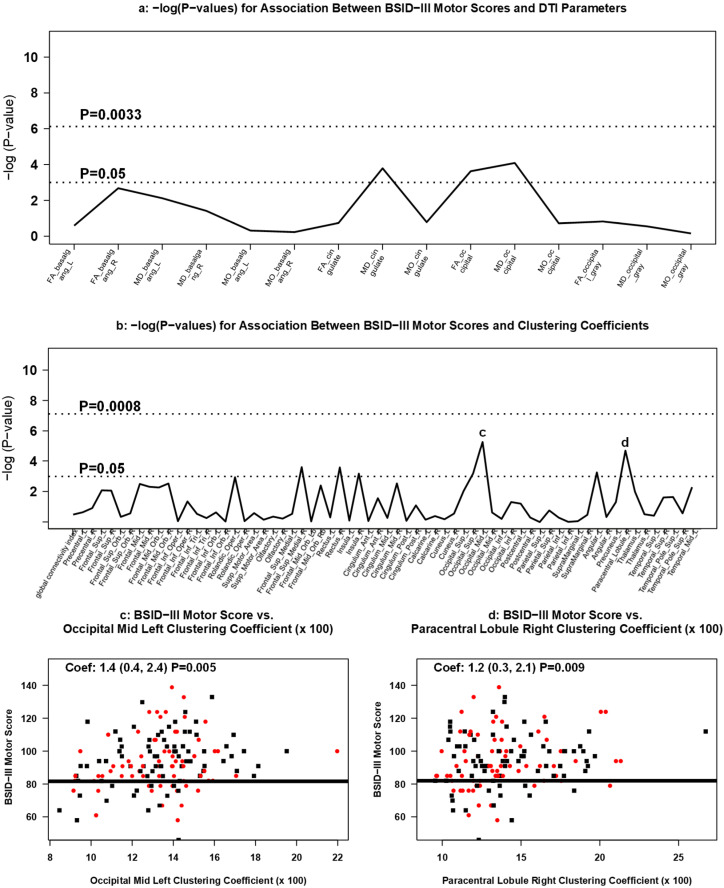
Association between DTI measures and BSID-III motor scores. Manhattan plots of [GEE-based Wald test] -log (*p*-values) for differences in (**a**) regional DTI diffusion values and (**b**) clustering coefficients by BSID-III motor scores. Panels (**c**,**d**) display scatterplots of BSID-III motor scores and statistically significant GEE-based associations with clustering coefficients. Red dots represent infants treated with Epo; black squares represent infants treated with placebo.

**Figure 7 brainsci-11-01360-f007:**
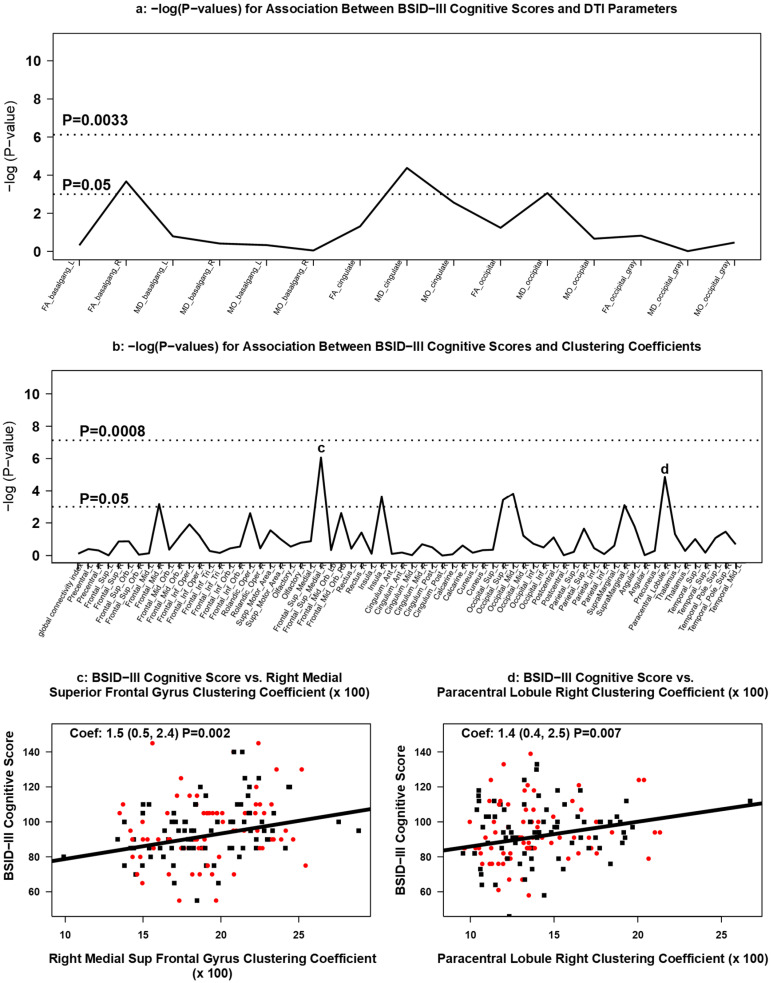
Association between DTI measures and BSID-III cognitive scores. Manhattan plots of [GEE-based Wald test] -log (*p*-values) for differences in (**a**) regional DTI diffusion values and (**b**) clustering coefficients by BSID-III cognitive scores. Panels (**c**,**d**) display scatterplots of BSID-III cognitive scores and statistically significant [GEE-based] associations in clustering coefficients. Red dots represent infants treated with Epo; black squares represent infants treated with placebo.

**Figure 8 brainsci-11-01360-f008:**
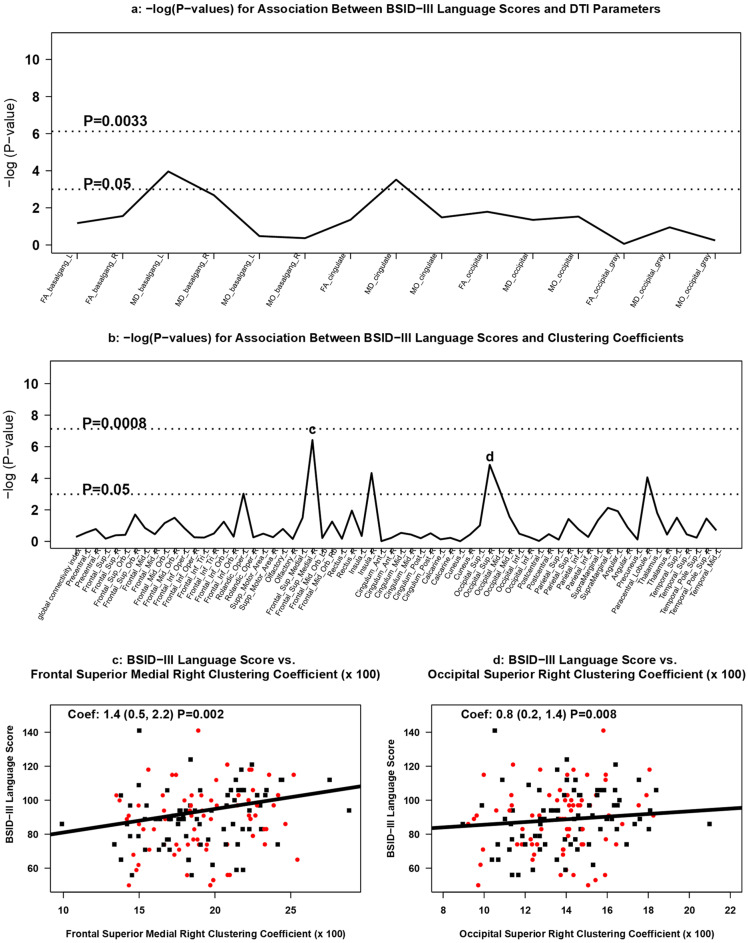
Association between DTI measures and BSID-III language scores. Manhattan plots of [GEE-based Wald test] -log (*p*-values) for differences in (**a**) regional DTI diffusion values and (**b**) clustering coefficients by BSID-III language scores. Panels (**c**,**d**) display scatterplots of BSID-III language scores and statistically significant [GEE-based] associations in clustering coefficients. Red dots represent infants treated with Epo; black squares represent infants treated with placebo.

**Table 1 brainsci-11-01360-t001:** Baseline demographics and clinical characteristics by treatment group.

	MRI Cohort			Non-MRI Cohort *
	Placebo	Epo	Overall	
Maternal demographics, *N* (%)	*N* = 101	*N* = 93	*N* = 194	*N* = 228
Age, mean (SD)	27.9 (6.2)	28.9 (6.6)	28.4 (6.4)	29.1 (6.2)
Race				
Hispanic	38 (38%)	27 (29%)	65 (34%)	51 (22%) **
White	62 (61%)	64 (69%)	126 (65%)	134 (59%)
Black	26 (26%)	23 (25%)	49 (25%)	77 (34%)
Other/Not reported	13 (13%)	6 (6%)	19 (10%)	17 (7.5%)
Education				
High School or less	45 (45%)	26 (28%)	71 (37%)	75 (33%)
Some college	23 (23%)	26 (28%)	49 (25%)	81 (36%)
College degree or greater	23 (23%)	25 (27%)	48 (25%)	49 (21%)
Not reported	10 (10%)	16 (17%)	26 (13%)	23 (11%)
Neonatal data at enrollment, *N* (%)				
Delivery complications	10 (10%)	13 (14%)	23 (12%)	33 (14%)
Antenatal steroids	92 (91%)	83 (89%)	175 (90%)	207 (91%)
Chorioamnionitis	16 (16%)	16 (17%)	32 (16%)	33 (14%)
Caesarean delivery	65 (64%)	60 (65%)	125 (64%)	161 (71%)
Delayed cord clamping	44 (58%)	49 (61%)	93 (60%)	67 (38%) ***
Female	56 (55%)	41 (44%)	97 (50%)	103 (45%)
Gestational age				
24 weeks	25 (25%)	15 (16%)	40 (20%)	57 (25%)
25 weeks	24 (24%)	23 (25%)	47 (24%)	56 (25%)
26 weeks	26 (26%)	27 (29%)	53 (28%)	58 (25%)
27 weeks	26 (26%)	28 (30%)	54 (28%)	57 (25%)
Mean (SD)	25.9 (1.2)	26.1 (1.1)	26.0 (1.1)	25.9 (1.1)
Multiple gestation	19 (19%)	24 (26%)	43 (22%)	60 (26%)
Infant weight (grams), mean (SD)	805.2 (176.3)	859.7 (177.6)	831.3 (178.6)	783.1 (183.4) ***
Apgar score at 5 min, mean (SD)	6.5 (1.8)	6.6 (1.9)	6.6 (1.9)	6.0 (2.1) ***
Epo level at birth, median (IQR)	*N* = 85 7.1 (4.2, 14.5)	*N* = 74 8.5 (4.8, 49.3)	*N* = 159 7.3 (4.4, 22.7)	*N* = 185 8.4 (4.2, 24.8)

* Among PENUT MRI recruitment sites. ** *p*-value for difference between MRI and non-MRI infants < 0.05. *** *p*-value for difference between MRI and non-MRI infants < 0.01. SD = standard deviation; IQR = interquartile range.

**Table 2 brainsci-11-01360-t002:** Complications and comorbidities between birth and 36 weeks’ PMA, and outcomes at age 2.

	MRI Cohort			Non-MRI Cohort *
	Placebo	Epo	Overall	
Postnatal markers of instability, *N* (%)	*N* = 101	*N* = 93	*N* = 194	*N* = 228
Necrotizing Enterocolitis (NEC)	6 (5.9%)	2 (2.2%)	8 (4.1%)	15 (6.6%)
Spontaneous Intestinal Perforation (SIP)	2 (2.0%)	1 (1.1%)	3 (1.5%)	11 (4.8%)
Sepsis	3 (3.0%)	3 (3.2%)	6 (3.1%)	28 (12%) **
Retinopathy of Prematurity (ROP)	8 (7.9%)	6 (6.5%)	14 (7.2%)	19 (8.3%)
Severe Intraventricular hemorrhage (IVH)	4 (5.9%)	2 (2.2%)	6 (3.1%)	36 (16%) §
Risk factors for NDI, *N* (%)				
Lowest ferritin in ng/mL (any time)				
<76	22/96 (23%)	61/89 (69%)	83/185 (45%)	75/200 (38%)
<40	6/96 (6.3%)	39/89 (44%)	45/185 (24%)	40/200 (20%)
Chronic lung disease (CLD)	42 (42%)	28 (30%)	70 (36%)	86 (38%)
Outcomes at Age 2, mean (SD)	*N* = 81	*N* = 73	*N* = 154	*N* = 184
BSID-III Cognitive	95.1 (15.8)	95.7 (18.6)	95.4 (17.2)	87.4 (16.1) §
BSID-III Motor	94.2 (15.9)	93.4 (16.7)	93.8 (16.2)	85.7 (17.4) §
BSID-III Language	89.8 (16.7)	88.2 (19.0)	89.0 (17.8)	85.7 (18.2)

* Among infants that survived through 36 weeks’ PMA at PENUT MRI recruitment sites. ** *p*-value for difference between MRI and Non-MRI infants < 0.01, [GEE-based Wald test] adjusted for GA at birth and treatment assignment. *p*-value for difference between Epo and placebo MRI infants < 0.001, [GEE-based Wald test] adjusted for GA at birth and treatment assignment. § *p*-value for difference between MRI and Non-MRI infants < 0.001, [GEE-based Wald test] adjusted for GA at birth and treatment assignment.

## Data Availability

De-identified individual participant data will be made available through the NINDS Data Archive: https://www.ninds.nih.gov/Current-Research/Research-Funded-NINDS/Clinical-Research/Archived-Clinical-Research-Datasets. The data will be de-identified and a limited access data set will be available through a request form on that page. Data dictionaries, in addition to study protocol, the statistical analysis plan, and the informed consent form will be included. The data will be made available upon publication of all PENUT Trial-related manuscripts to researchers who provide a methodologically sound proposal for use in achieving the goals of the approved proposal.

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
