# Peer review of "Diffusion Tensor Imaging Changes Do Not Affect Long-Term Neurodevelopment following Early Erythropoietin among Extremely Preterm Infants in the Preterm Erythropoietin Neuroprotection Trial"

_brainsci, 2021, doi:10.3390/brainsci11101360_

Round 1

Reviewer 1 Report

Re: brainsci-1410518 by Janessa Law , Bryan A Comstock , Todd Richards , Chris M Traudt , Thomas R. Wood , Dennis E. Mayock , Patrick J. Heagerty , Sandra E. Juul. This manuscript reports on the relationship between dti-MRI and neurodevelopmental (Bayley-III) outcomes after rEpo therapy in extreme preterm infants, and is a follow-up report to their well-publicized PENUT trial published in N Engl J Med in 2020. PENUT and its associated studies have high clinical relevance, being the first RCT trial on preterm rEpo neuroprotection. The authors are a highly accomplished, well-known clinico-research team, with an extensive, track record on rEpo. Overall, the paper is well-written, minus several minor remarks.

-Methodology needs to include more detail on the parent PENUT trial. How and when was rEpo given, and how many infants were enrolled in the main trial? Appreciate this has been published, but a brief description is desirable to contextualize these sub-analyses. How many infants ended up being used for analyses in the 24-25 week and 26-27 week groups. Please include in figure 1.

- The authors have just published 2 key follow-up PENUT papers that have bearing on the current results. These are not mentioned in this manuscript. In these papers, a correlation is shown between iron supplementation and MRI findings, and 2Y Bayley scores. Did infants in this cohort receive iron supplements? If so, how many infants, what dosing, and was there difference between the treatment groups? The current data show minor MRI-dti changes, whereas a number of infants had mild lesions on conventional MRI. Was there overlap in infants between these studies? The relevance of these previous findings to the current data, if any, should be discussed.

- Line 409 - FA and MD vs GA; MD was not significant in results.

- Line 435 - inconsistency, full-stop before references.

- Figure 7 has poor quality fonts, and is hard to read.

Dennis E Mayock 1, Semsa Gogcu 2, Mihai Puia-Dumitrescu 3, Dennis W W Shaw 4, Jason N Wright 4, Bryan A Comstock 5, Patrick J Heagerty 5, Sandra E Juul 3, PENUT Trial Consortium. Association between Term Equivalent Brain MRI and 2 Year Outcomes in Extremely Preterm Infants: A Report from the PENUT Trial Cohort J Pediatr. 2021 Aug 26;S0022-3476(21)00825-8.

Kendell R German 1, Phuong T Vu 2, Bryan A Comstock 2, Robin K Ohls 3, Patrick J Heagerty 2, Dennis E Mayock 4, Michael Georgieff 5, Raghavendra Rao 5, Sandra E Juul 4, PENUT Consortium. Enteral Iron Supplementation in Infants Born Extremely Preterm and its Positive Correlation with Neurodevelopment; Post Hoc Analysis of the Preterm Erythropoietin Neuroprotection Trial Randomized Controlled Trial. J Pediatr . 2021 Jul 27;S0022-3476(21)00686-7.

Author Response

Re: brainsci-1410518 by Janessa Law , Bryan A Comstock , Todd Richards , Chris M Traudt , Thomas R. Wood , Dennis E. Mayock , Patrick J. Heagerty , Sandra E. Juul. This manuscript reports on the relationship between dti-MRI and neurodevelopmental (Bayley-III) outcomes after rEpo therapy in extreme preterm infants, and is a follow-up report to their well-publicized PENUT trial published in N Engl J Med in 2020. PENUT and its associated studies have high clinical relevance, being the first RCT trial on preterm rEpo neuroprotection. The authors are a highly accomplished, well-known clinico-research team, with an extensive, track record on rEpo. Overall, the paper is well-written, minus several minor remarks.

Thank you for the kind words and acknowledgement of the PENUT team. We have provided point-by-point responses below as well as highlighted relevant changes in the manuscript.

-Methodology needs to include more detail on the parent PENUT trial. How and when was rEpo given, and how many infants were enrolled in the main trial? Appreciate this has been published, but a brief description is desirable to contextualize these sub-analyses. How many infants ended up being used for analyses in the 24-25 week and 26-27 week groups. Please include in figure 1.

We agree this information is beneficial for context and have included total infants enrolled in the PENUT Trial to the Results section (Page 6 Line 253) as well as other relevant information such as Epo treatment timing to the Methods section (Page 2 Line 81), and transfusion and hematocrit data to the Discussion (Page 20 Lines 680-684). The number of infants in each group by gestational age was provided in Table 1, but we were happy to add this information to Figure 1 as well (Page 6 Line 260).

- The authors have just published 2 key follow-up PENUT papers that have bearing on the current results. These are not mentioned in this manuscript. In these papers, a correlation is shown between iron supplementation and MRI findings, and 2Y Bayley scores. Did infants in this cohort receive iron supplements? If so, how many infants, what dosing, and was there difference between the treatment groups? The current data show minor MRI-dti changes, whereas a number of infants had mild lesions on conventional MRI. Was there overlap in infants between these studies? The relevance of these previous findings to the current data, if any, should be discussed.

We found in a separate publication that iron status as investigated by enteral dosing is linked to neurodevelopmental outcomes and that reference has been added to the paper, thank you that observation (Page 5 Line 222 and Page 19 Line 674).

The PENUT Trial protocol defined the procedure for iron supplementation for all infants enrolled: "All infants began receiving iron supplementation when their enteral feeding volume had reached 60 ml per kilogram per day and they were at least 7 days old. Infants initially received 3 mg per kilogram per day of enteral iron. The dose was increased to 6 mg per kilogram per day when infants reached a feeding volume of 100 to 120 ml per kilogram per day. Serum ferritin or the ratio of zinc protoporphyrin to heme was assessed on days 14 and 42, and the dose of supplemental iron was adjusted accordingly. Infants who did not receive enteral feedings received parenteral iron (1.5 mg per kilogram twice a week, adjusted on the basis of the ratio of zinc protoporphyrin to heme or serum ferritin values).” (Juul 2020)

Thus, all infants received some level of iron supplementation. Infants in the Epo group required a higher dose of both enteral and IV iron (provided in the primary paper supplemental material and added to the manuscript on Page 19, Line 675) likely due to increased erythropoiesis. Despite the increased dose, more Epo- than placebo-treated infants had evidence of moderate or severe iron deficiency during their hospitalization. We chose to include ferritin levels, a surrogate of iron sufficiency, in this manuscript given the association we previously found between iron status and neurodevelopmental outcome (explanation added to Page 5 Line 222).

We appreciate the inquiry into examining higher order interactions between infant characteristics, MRI injury scoring, and DTI parameters. In the analyses presented in the manuscript, we examined all DTI parameters and clustering coefficients individually to address two sets of questions: first, did any of these measurements differ by treatment group or gestational age at birth?, and second, were any of these measures prognostic of outcomes at age 2? The analyses presented were specified a priori as a part of the study protocol and statistical analysis plan, with one exploratory post-hoc analysis of treatment by gestational age interactions. We plan to investigate this question further as a broader set of exploratory analyses in a separate manuscript related to prediction of outcomes at age 2. Similarly, comparison of gross MRI injury to DTI findings was outside the scope of this paper but will be included in a separate planned paper.

Juul, S.E., et al., A Randomized Trial of Erythropoietin for Neuroprotection in Preterm Infants. N Engl J Med, 2020. 382(3): p. 233-243.

- Line 409 - FA and MD vs GA; MD was not significant in results and - Line 435 - inconsistency, full-stop before references.

While these edits are appreciated and we would like to make them, it seems that our line numbers may be different than the ones you mentioned. Can you clarify which sentences you are referring to so that we can make these changes?

- Figure 7 has poor quality fonts, and is hard to read.

Thank you for bringing this to our attention. Figure 7 as well as Figures 6 and 8 have been updated for clarity (Pages 16-18).

Reviewer 2 Report

This study should have received attention by medical doctors who work in this field. It seems good to show DTI findings of the NEJM published clinical trial because it helps understanding about the negative finding of the trial.

I have questions and wish to be updated regarding the following issues.

Methods

1) Final voxel size: In the preprocessing procedure, did the authors resampled the images to the same voxel sizes across the sites and magnets? E.g., 1mm isometric.

2) Cross-site comparability: Between Siemens and Philips, the authors reported average FA values with a standard deviation and claimed the values were similar. However, a more quantitative comparison would be useful for readers to get a sense of comparability between the diffusion-weighted images and tensor images obtained using two different magnets and protocols (# of gradient directions, voxel sizes, etc). Also, instead of mean FA values, SNR should be a better metric for cross-site, cross-protocol comparison. Please consider it. 

3) Design matrix for DTI Randomise permutation: Detailed descriptions of the statistical models used to discover “significant group differences” in the whole brain analysis are required.

Any covariates? 

How many permutations were used?

What was the mask used in the whole-brain analysis, e.g., skeletonised white matter mask, or the whole brain mask including GM and WM?

Multiple comparison correction for FA, MO, MD, L1, L2, L3 considered? (The authors did this in the regression analysis with the extracted values though) 

4) Detailed descriptions of the probabilistic tractography are necessary. Otherwise, the reproducibility of the results may be questionable. 

Results

1) The authors mentioned fewer blood transfusion of the Epo-treated patients in Discussion. However, Results section does not provide this data. Also, exact time of Epo treatment and the time of MRI should be clarified.

2) In Table 1, Row of Race and Hispanic seems to be switched. And what does “HS” stand for?

3) If specific grouping is possible using DTI (E.g., severe cases with low FA values), how about analyze their outcome at 2 years age?

Discussion

1) Duration between Epo treatment and MRI taking time could be too short to see the effect of Epo. More discussion is needed on this issue.

2) According to the result of increased white matter MD in Epo-treated patients, Epo treatment during NICU care may be interpreted as even harmful. There can be other factors that conceal neuroprotective effect of Epo. Use of steroid could be one of it. And, by conducting further study, we can find specific subgroup that can get benefit by Epo administration. Please add possible directions for further study afterwards.

Author Response

This study should have received attention by medical doctors who work in this field. It seems good to show DTI findings of the NEJM published clinical trial because it helps understanding about the negative finding of the trial.

Thank you for your review and taking to the time to help us improve our summary of DTI findings in the PENUT cohort. We have provided point-by-point responses below as well as highlighted relevant changes in the manuscript.

I have questions and wish to be updated regarding the following issues.

Methods

1) Final voxel size: In the preprocessing procedure, did the authors resampled the images to the same voxel sizes across the sites and magnets? E.g., 1mm isometric.

Yes. In the procedure that describes co-registration, all subjects are transformed into the same space so that group voxelwise analysis can be formed. This explanation was added to the manuscript (Page 3 Line 136).

2) Cross-site comparability: Between Siemens and Philips, the authors reported average FA values with a standard deviation and claimed the values were similar. However, a more quantitative comparison would be useful for readers to get a sense of comparability between the diffusion-weighted images and tensor images obtained using two different magnets and protocols (# of gradient directions, voxel sizes, etc). Also, instead of mean FA values, SNR should be a better metric for cross-site, cross-protocol comparison. Please consider it. 

We agree, SNR would be a good alternative metric for cross-site and cross-protocol comparison. The standard measure of diffusion across the majority of studies is FA and we feel that FA provides an adequate measure for comparison between scanners. Although we would not be able to perform this analysis rapidly, we appreciate the input and plan to pursue this comparison in the future.

3) Design matrix for DTI Randomise permutation: Detailed descriptions of the statistical models used to discover “significant group differences” in the whole brain analysis are required.

Any covariates? Only treatment group comparisons were performed for identification of areas with significant white matter differences. We corrected for multiple covariates including gestation age, sex, and scanner type for our final statistical analyses (Page 5 Line 227).

How many permutations were used? 1000 permutations were used (Page 4 Line 152).

What was the mask used in the whole-brain analysis, e.g., skeletonised white matter mask, or the whole brain mask including GM and WM?

ANTs provided a whole brain template including GM and WM (Page 4 Line 143). We acknowledge there is some debate regarding the ability to perform gray and white matter segmentation in infants. We chose to use ANTs in an effort to overcome the limitations imposed by the immature white matter of the premature brain.

Multiple comparison correction for FA, MO, MD, L1, L2, L3 considered? (The authors did this in the regression analysis with the extracted values though).

Initial analysis with FSL Randomise corrected for multiple voxels, and multiple comparison with correction for FA, MO, MD, and exclusion of L1, L2, and L3 as the correlation coefficient between these gradients and MD in all ROIs was close to 1, was performed for the final analysis (Page 5 Lines 231-235).

4) Detailed descriptions of the probabilistic tractography are necessary. Otherwise, the reproducibility of the results may be questionable. 

We utilized FSLs probtrackx2 software for connectivity analysis. The following commands used are provided below and were added to the Appendix (Pages 25-26, Lines 896-940).

Bedpost:

#!/bin/bash

cd /mnt/neuroimaging2/todd/penut/prepare_for_dtiscript

list=`ls -d group2_subgr2*`

for afolder in ${list}

do

echo working on ${afolder}

cd /mnt/neuroimaging2/todd/penut/prepare_for_dtiscript/${afolder}

mkdir bedpost

cp out/mc_*.nii.gz bedpost/data.nii.gz

cp rearranged_bvals.txt bedpost/bvals

cp out/bvec_mc.txt bedpost/bvecs

cp mask1.nii.gz bedpost/nodif_brain_mask.nii.gz

bedpostx bedpost -n 2

Probtackx2:

cd /mnt/neuroimaging2/todd/penut/prepare_for_dtiscript

list=`ls -d group*inter*`

for afolder in ${list}

do

echo working on ${afolder}

cd /mnt/neuroimaging2/todd/penut/prepare_for_dtiscript/${afolder}

flirt -in /mnt/neuroimaging2/todd/penut/connectome/atlas/AVG3-0Months3T_brain_t2w114.nii.gz -ref mask1 -out braintos0 -omat braintos0.mat -bins 256 -cost corratio -searchrx -90 90 -searchry -90 90 -searchrz -90 90 -dof 12  -interp trilinear

flirt -in /mnt/neuroimaging2/todd/penut/connectome/atlas/ANTS3-0Months3T_brain_AAL_atlas_detailed114.nii.gz  -ref mask1 -out atlastos0 -applyxfm -init braintos0.mat -interp nearestneighbour

listseeds='1 2 3 4 5 6 7 8 9 10 11 12 13 14 15 16 17 18 19 20 21 22 23 24 25 26 27 28 29 30 31 32 33 34 35 36 43 44 45 46 49 50 51 52 53 54 57 58 59 60 61 62 63 64 65 66 67 68 69 70 77 78 81 82 83 84 85 86 87 88 89 90 91 92'

rm seed*

for i in $listseeds

do

echo $i;

result1=$(echo "(0.5 +${i})" | bc -l )

result2=$(echo "(-0.5 +${i})" | bc -l )

echo $result1 $result2

fslmaths atlastos0 -thr ${result2} -uthr ${result1} -dilM seed${i}

done

#each seed region

probtrackx2 --network -x listseeds.txt  -l --onewaycondition --omatrix1 -c 0.2 -S 1000 --steplength=0.5 -P 1000 --fibthresh=0.01 --distthresh=0.0 --sampvox=0.0 --forcedir --opd -s bedpost.bedpostX/merged -m bedpost.bedpostX/nodif_brain_mask  --dir=probtrackoutput

Results

1) The authors mentioned fewer blood transfusion of the Epo-treated patients in Discussion. However, Results section does not provide this data. Also, exact time of Epo treatment and the time of MRI should be clarified.

Detailed investigation regarding Fe levels and the number of transfusions in the Epo-treated vs placebo-treated animals has been previously published: Juul, S.E., et al., Effect of High-Dose Erythropoietin on Blood Transfusions in Extremely Low Gestational Age Neonates: Post Hoc Analysis of a Randomized Clinical Trial. JAMA Pediatr, 2020. 174(10): p. 933-943. We agree this information is beneficial for context and have included total infants enrolled in the PENUT Trial to the Results section (Page 6 Line 253) as well as other relevant information such as Epo treatment timing to the Methods section (Page 2 Line 81), and transfusion and hematocrit data to the Discussion (Page 20 Lines 680-684).

2) In Table 1, Row of Race and Hispanic seems to be switched. And what does “HS” stand for?

Thank you for catching this error. The table has been edited and “HS” has been clarified to “High School” (Page 7 Line 296).

3) If specific grouping is possible using DTI (E.g., severe cases with low FA values), how about analyze their outcome at 2 years age?

We appreciate the inquiry into examining higher order interactions between infant characteristics and DTI parameters. In the analyses presented in the manuscript, we examined all DTI parameters and clustering coefficients individually to address two sets of questions: first, did any of these measurements differ by treatment group or gestational age at birth?, and second, were any of these measures prognostic of outcomes at age 2? The analyses presented were specified a priori as a part of the study protocol and statistical analysis plan, with one exploratory post-hoc analysis of treatment by gestational age interactions. We plan to investigate this question further as a broader set of exploratory analyses in a separate manuscript related to prediction of outcomes at age 2.

Discussion

1) Duration between Epo treatment and MRI taking time could be too short to see the effect of Epo. More discussion is needed on this issue.

Thank you for including this observation in your review. While the first dose of Epo was given within the first 24 hours of life, the final dose was not received until 32 weeks and 6 days post-menstrual age. It is absolutely possible that the white matter effects would not have been visible by 36-37 weeks post-menstrual age, which is when the scans were performed. Additionally, it has been well documented that FA values continue to increase as infants approach full term, thus additional white matter tract maturation over time maybe have facilitated better FA signal detection. These observations a well as a reference (below) regarding the timeline between FA and gestational age have been added to the limitations (Page 21 Lines 735-740).

Wilson, S., et al., Development of human white matter pathways in utero over the second and third trimester. Proc Natl Acad Sci U S A, 2021. 118(20).

2) According to the result of increased white matter MD in Epo-treated patients, Epo treatment during NICU care may be interpreted as even harmful. There can be other factors that conceal neuroprotective effect of Epo. Use of steroid could be one of it. And, by conducting further study, we can find specific subgroup that can get benefit by Epo administration. Please add possible directions for further study afterwards.

We agree that several confounding factors such as steroids (manuscript in process) and exposure to opiates and/or benzodiazepines (please see reference to our recently published article below) could mask the neuroprotective effect of Epo (Page 19 Line 665). As these exposures vary significantly in the clinical setting, early identification of subgroups can be challenging. However, given that we speculate iron deficiency, and not Epo itself, may have contributed to the increased white matter MD values seen in our extremely preterm Epo-treated patients, ensuring adequate iron supplementation in these infants is an actionable intervention that deserves further study. Investigation into outcomes of iron sufficient versus iron deficient infants in this cohort is ongoing.

Puia-Dumitrescu, M., et al., Assessment of 2-Year Neurodevelopmental Outcomes in Extremely Preterm Infants Receiving Opioids and Benzodiazepines. JAMA Netw Open, 2021. 4(7): p. e2115998.

Reviewer 3 Report

In this manuscript, Law et al. report the results of MRI diagnostics in a subgroup of patients included in the recent phase III randomized controlled trial (RCT) on the effect of early recombinant erythropoietin (rEpo) on brain development in very preterm infants (PENUT). MRI scans with diffusion tensor imaging were performed at 36 weeks postmenstrual age in 101 (out of 229) placebo-treated and 93 (out of 240) rEpo-treated infants. The authors have chosen two regions of interest for the data analysis: a) the cingulate white matter near the cingulate gyrus and the occipital white matter, and b) bilateral basal ganglia and occipital regions in the grey matter. The results indicated the following: (i) a lower mean diffusivity in both cingulate and occipital white matter of placebo-treated infants, (ii) no other differences in diffusion measures in the analyzed (or any other [see page 8 line 283: reported as data not shown]) areas of white or grey matter, and (iii) fractional anisotrophy and mode measures did also not differ between the two study groups. Perhaps most importantly, neither white nor grey matter diffusion tensor imaging were associated with Bayley Scales of Infant and Toddler Development (BSID-III) at the age of 2 years. Although different analyses of coefficient clustering showed difference associations to various scores in the BSID-III, none of these findings reached statistical significance in multiple variant testing.

General comment:

The analysis of MRI scans among patients of the PENUT trial is a priori of high interest. Although the authors discuss several limitations of their study, additional questions and concerns need to be addressed.

Major critique:

  • Overall, the MRI scans, performed at 36 weeks postmenstrual age, do not indicate a significant difference in brain development of very preterm infants treated with initial high- and subsequent low-dose rEpo for neuroprotection. This structural analysis is also not associated with neurodevelopmental outcome at 2-years of age. This is in line with the initial report on clinical outcome measure of this RCT, published in the NEJM 2020. This should stated more clearly in the title and abstract of the manuscript. The speculations the mean diffusivity (MD) changes in Epo-treated infants and fractional anisotrophy (FA) in the most preterm infants ‘may represent delayed neural maturation’ is confusing. In line, the final statement that ‘structural connectivity when described by clustering coefficients may indicate long-term risk of NDL’ (neurodevelopmental delay) should be avoided, since it could be interpreted as negative side effect of early rEpo treatment.
  • Previous data of a retrospective case-control study by Neubauer AP et al. (2010) indicated that rEpo treatment improved neurodevelopmental outcome only in the group of preterm infants with severe intraventricular hemorrhage (IVH, stage III/IV). Since the MRI cohort of the PENUT trial had significantly fewer infants with culture proven sepsis and grade III/IV IVH (p.8 l. 252 ff), the level of significance in MRI findings could be too low for indicating beneficial effects of rEpo. This aspect must be considered in a revised manuscript.
  • In general, the definition of the region of interest for the analysis and interpretation of MRI scans is of significant major concern (p4, l146-151). Regardless of the difference of age at the time of performing the MRI (36 weeks pc in the PENUT vs. term in the Swiss Neuroprotection trial), the author should seriously attempt to reach comparability between both RCTs by the addition of scores for white matter injury (WMI) and grey matter injury (GMI) as done by Leuchter RHV et al. (JAMA, 2014). These parameters are easy to obtain and to evaluate (e.g. for WMI: periventricular white matter loss, cystic abnormalities, ventricular dilatation, thinning of the callosum, or for GMI: cortical abnormality, gyral maturation and subarachnoidal space). In this reviewer’s understanding, such analysis is crucial for the revision of the manuscript.
  • It remains very speculative of whether iron-deficiency contributes to the increased white matter mean diffusivity in the extremely preterm rEpo-treated infants (p15, L.382-393). The data do not support this speculation, in particular, since differences in serum ferritin levels are not significant. This parameter may not reach enough value as ferritin acts like an acute phase response protein to more general health conditions. Instead, analyzing hepcidin concentrations would be of interest.
  • Besides the consideration of the analysis by Leuchter et al. (JAMA 2014), the results need to be discussed more thoroughly in the context of other MRI examinations performed in order to follow-up the putative neuroprotective effects of rEpo in preterm infants: (i) Yang SS et al., Chin J Contemp Pediatr 2018, (ii) Philipps J et al., Pediatr Res 2017, (iii) Gasparovic C et al., Pediatr Radiol 2018.

Minor points:

  • Introduction: Paragraph introduction (p.2, l.51-63) on limitations in assessing the association between MRI findings and neurodevelopment outcome is redundant. Instead, one or two references could be added to the paragraph in which the authors discuss the limitations of their study (p.16, l.433-444).
  • Discussion (p15, l.379): Reference 66 (O’Gorman RL et al.) instead of 67 would be correct, would it not?

Author Response

In this manuscript, Law et al. report the results of MRI diagnostics in a subgroup of patients included in the recent phase III randomized controlled trial (RCT) on the effect of early recombinant erythropoietin (rEpo) on brain development in very preterm infants (PENUT). MRI scans with diffusion tensor imaging were performed at 36 weeks postmenstrual age in 101 (out of 229) placebo-treated and 93 (out of 240) rEpo-treated infants. The authors have chosen two regions of interest for the data analysis: a) the cingulate white matter near the cingulate gyrus and the occipital white matter, and b) bilateral basal ganglia and occipital regions in the grey matter. The results indicated the following: (i) a lower mean diffusivity in both cingulate and occipital white matter of placebo-treated infants, (ii) no other differences in diffusion measures in the analyzed (or any other [see page 8 line 283: reported as data not shown]) areas of white or grey matter, and (iii) fractional anisotrophy and mode measures did also not differ between the two study groups. Perhaps most importantly, neither white nor grey matter diffusion tensor imaging were associated with Bayley Scales of Infant and Toddler Development (BSID-III) at the age of 2 years. Although different analyses of coefficient clustering showed difference associations to various scores in the BSID-III, none of these findings reached statistical significance in multiple variant testing.

Thank you for your review and taking to the time to help us improve our summary of DTI findings in the PENUT cohort. We have provided point-by-point responses below as well as highlighted relevant changes in the manuscript.

General comment:

The analysis of MRI scans among patients of the PENUT trial is a priori of high interest. Although the authors discuss several limitations of their study, additional questions and concerns need to be addressed.

Major critique:

Overall, the MRI scans, performed at 36 weeks postmenstrual age, do not indicate a significant difference in brain development of very preterm infants treated with initial high- and subsequent low-dose rEpo for neuroprotection. This structural analysis is also not associated with neurodevelopmental outcome at 2-years of age. This is in line with the initial report on clinical outcome measure of this RCT, published in the NEJM 2020. This should stated more clearly in the title and abstract of the manuscript. The speculations the mean diffusivity (MD) changes in Epo-treated infants and fractional anisotrophy (FA) in the most preterm infants ‘may represent delayed neural maturation’ is confusing. In line, the final statement that ‘structural connectivity when described by clustering coefficients may indicate long-term risk of NDL’ (neurodevelopmental delay) should be avoided, since it could be interpreted as negative side effect of early rEpo treatment.

We appreciate this recommendation to clarify and focus our findings. The title of the manuscript as well as the conclusion of the abstract have been edited to better relate to the primary study report (Page 1,Lines 23-25).

Previous data of a retrospective case-control study by Neubauer AP et al. (2010) indicated that rEpo treatment improved neurodevelopmental outcome only in the group of preterm infants with severe intraventricular hemorrhage (IVH, stage III/IV). Since the MRI cohort of the PENUT trial had significantly fewer infants with culture proven sepsis and grade III/IV IVH (p.8 l. 252 ff), the level of significance in MRI findings could be too low for indicating beneficial effects of rEpo. This aspect must be considered in a revised manuscript.

We appreciate the inquiry into examining higher order interactions between infant characteristics and DTI parameters. In the analyses presented in the manuscript, we examined all DTI parameters and clustering coefficients individually to address two sets of questions: first, did any of these measurements differ by treatment group or gestational age at birth?, and second, were any of these measures prognostic of outcomes at age 2? The analyses presented were specified a priori as a part of the study protocol and statistical analysis plan, with one exploratory post-hoc analysis of treatment by gestational age interactions. Evaluating clinical risk factors for DTI changes was outside the scope of this manuscript.

However, we find this topic of particular interest and are happy to discuss it further. While we acknowledge the findings of Neubauer et al, their retrospective study only included 10 infants born in a different era of Neonatal care (1993-1998) with severe IVH (Grade 3 or 4) from which to base their statistical analysis. We have recently published the largest contemporary study on IVH and neurodevelopmental outcomes in extremely preterm infants based on a secondary analysis of the prospective, blinded, and randomized PENUT Trial (Law 2021). We did not find any effect of Epo on the neurodevelopmental outcome in infants with Grade 3 or Grade 4 hemorrhage. Additionally, the data demonstrates that Grade 3 and Grade 4 hemorrhages result in drastically different neurodevelopmental outcomes. Without detailed information regarding the number of infants in each group of the Neubaeuer study with Grade 3 versus Grade 4 hemorrhage, it is difficult to know if the effect can be attributed to Epo or to unbalanced injury grades between groups (i.e. more Grade 3 hemorrhages in the Epo group and more Grade 4 hemorrhages in the placebo group could explain their outcome). We are keen to evaluate more closely the effect of IVH (and many other clinical characteristics) on DTI parameters in an upcoming manuscript.

Law JB, Wood TR, Gogcu S, et al. Intracranial Hemorrhage and 2-Year Neurodevelopmental Outcomes in Infants Born Extremely Preterm. The Journal of Pediatrics. 2021 Jul. DOI: 10.1016/j.jpeds.2021.06.071. PMID: 34217769.

In general, the definition of the region of interest for the analysis and interpretation of MRI scans is of significant major concern (p4, l146-151). Regardless of the difference of age at the time of performing the MRI (36 weeks pc in the PENUT vs. term in the Swiss Neuroprotection trial), the author should seriously attempt to reach comparability between both RCTs by the addition of scores for white matter injury (WMI) and grey matter injury (GMI) as done by Leuchter RHV et al. (JAMA, 2014). These parameters are easy to obtain and to evaluate (e.g. for WMI: periventricular white matter loss, cystic abnormalities, ventricular dilatation, thinning of the callosum, or for GMI: cortical abnormality, gyral maturation and subarachnoidal space). In this reviewer’s understanding, such analysis is crucial for the revision of the manuscript.

The Swiss Epo Neuroprotection Trial has published several articles related to the effect of Epo on MRI findings. Their first imaging paper published by Leuchter in JAMA in 2014 evaluated WM and GM injury according to the scoring system published by Woodward et al that established a qualitative approach to injury scoring. This is similar and directly comparable to the scoring system we utilized in a recently published paper using a modified Kidokoro score in which we found no effect of Epo treatment on MRI brain injury compared to placebo-treated infants. We feel this paper is directly in line with your request and can be found here: https://pubmed.ncbi.nlm.nih.gov/34454953/.

When referencing the The Swiss Epo Neuroprotection Trial in this manuscript, we were specifically referring to their paper by O’Gorman evaluating the effect of Epo as assessed by DTI and tract-based spatial statistics. They found significantly higher mean FA values in Epo-treated infants which was consistent with their qualitative findings in the Leuchter paper above. Similarly, our DTI analysis is consistent with the findings of our qualitative MRI paper – that Epo did not affect imaging findings. We believe the variation in dosing, gestational age at birth, and timing of imaging may have all played a role in the different findings between our two studies. Exposure to medications such as post-natal steroids, opiates, and benzodiazepines are detrimental to neurodevelopment and may have masked the neuroprotective effect of Epo in our cohort (Puia Dumitrescu 2021).

Leuchter RH, et al., Association between early administration of high-dose erythropoietin in preterm infants and brain MRI abnormality at term-equivalent age. JAMA. 2014 Aug 27;312(8):817-24. doi: 10.1001/jama.2014.9645. PMID: 25157725.

Woodward, L.J., et al., Neonatal white matter abnormalities an important predictor of neurocognitive outcome for very preterm children. PLoS One, 2012. 7(12): p. e51879.

Mayock DE, et al., Association between Term Equivalent Brain MRI and 2 Year Outcomes in Extremely Preterm Infants: A Report from the PENUT Trial Cohort. J Pediatr. 2021 Aug 26:S0022-3476(21)00825-8.

O'Gorman, R.L., et al., Tract-based spatial statistics to assess the neuroprotective effect of early erythropoietin on white matter development in preterm infants. Brain, 2015. 138(Pt 2): p. 388-97.

Puia-Dumitrescu, M., et al., Assessment of 2-Year Neurodevelopmental Outcomes in Extremely Preterm Infants Receiving Opioids and Benzodiazepines. JAMA Netw Open, 2021. 4(7): p. e2115998.

It remains very speculative of whether iron-deficiency contributes to the increased white matter mean diffusivity in the extremely preterm rEpo-treated infants (p15, L.382-393). The data do not support this speculation, in particular, since differences in serum ferritin levels are not significant. This parameter may not reach enough value as ferritin acts like an acute phase response protein to more general health conditions. Instead, analyzing hepcidin concentrations would be of interest.

We agree that iron-deficiency’s role in increased WM MD in Epo-treated infants is speculative. However, we respectfully direct you to Table 2 where indeed there were significantly more infants in the Epo-treated group than the placebo-treated group that had low ferritin levels (p < 0.001, GEE-based Walk test adjusted for GA at birth and treatment assignment). As you mentioned, ferritin acts like an acute phase reactant thus we would posit that higher ferritin levels would be associated with inflammatory insults while low ferritin levels (like those seen in the Epo cohort) are more representative of low iron stores.

We have recently demonstrated that iron status as investigated by enteral dosing is linked to neurodevelopmental outcomes and that reference has been added to the paper (Page 5 Line 222 and Page 19 Line 674). In the meantime, we are completing a separate paper evaluating the role of hepcidin as we concur this parameter is less affected by confounders, and we are investigating the role of iron sufficiency versus iron deficiency in a broader set of exploratory analyses in a separate manuscript related to prediction of outcomes at age 2. We hope that this rich dataset will help to form a more complete picture of the role of iron in the neurodevelopment of extremely preterm infants.

German, K.R., et al., Enteral Iron Supplementation in Infants Born Extremely Preterm and its Positive Correlation with Neurodevelopment; Post Hoc Analysis of the Preterm Erythropoietin Neuroprotection Trial Randomized Controlled Trial. J Pediatr, 2021.

Besides the consideration of the analysis by Leuchter et al. (JAMA 2014), the results need to be discussed more thoroughly in the context of other MRI examinations performed in order to follow-up the putative neuroprotective effects of rEpo in preterm infants: (i) Yang SS et al., Chin J Contemp Pediatr 2018, (ii) Philipps J et al., Pediatr Res 2017, (iii) Gasparovic C et al., Pediatr Radiol 2018.

We appreciate this recommendation and perspective. Gasparovic C et al evaluated MRS changes which is not directly applicable to this study, but we have included the work by Phillips et al and Yang et al to our Discussion (Page 19 Lines 660-665).

Minor points:

Introduction: Paragraph introduction (p.2, l.51-63) on limitations in assessing the association between MRI findings and neurodevelopment outcome is redundant. Instead, one or two references could be added to the paragraph in which the authors discuss the limitations of their study (p.16, l.433-444).

While we recognize your point, we’d like to clarify that this paragraph is intended not to discuss a limitation of the study but to introduce the concept of connectivity, and specifically clustering coefficients, just as we introduced the concepts of fractional anisotropy and mean diffusivity in the paragraph beforehand. We anticipate that this article may be of interest to clinicians who are not entirely familiar with the concepts and terminology of Diffusion Tensor Imaging and thus we feel these paragraphs provide a framework within which clinicians can understand the relationship between imaging and function.

Discussion (p15, l.379): Reference 66 (O’Gorman RL et al.) instead of 67 would be correct, would it not?

Thank you very much for finding this error.

Reviewer 4 Report

This is a well-written paper using DTI to assess the effect of EPO in preterm infants. The method is well written, the statistical model is strict, and the results are well presented. But I have several suggestions for the paper:

  1. I really appreciate the researchers consider the site and scanner difference, when designing the study. When saying Siemens and Philips values are similar (line 99), could you provide the mean of two scanners, and p-value of the comparison? 
  2. Co-registration of infant's brain is a difficult job. Could you provide the registration result to convince us the registration is accurate? Otherwise, the detected regions may be biased.
  3. The definition of five ROIs. Please clarify the details about how to get these regions. AAL has lots of regions, why only these regions are included? Is there a threshold of voxel count? And, please also provide the accurate name of these regions. AAL has anterior/mid/posterior cingulate gyrus, as well as lots of occipital regions.
  4. Clustering coefficient is just one parameter to represent information transfer ability within a graph. There are other parameters, such as local/global efficiency, small worldness. These are also widely used in previous papers. Why only considered clustering coefficient in this study?  Specifically, clustering coefficient is associated with local efficiency. In other word, only using clustering coefficient to represent information transfer efficiency is relatively biased.
  5. Setting FC matrix at 10% sparsity is arbitrary. A strict way is to test the robustness of the findings at different sparsities (i.e. 5%, 10%,15%,20%,25%,30%). Or use the AUC (1%-50% sparsity), instead of clustering coefficient from a specific sparsity, in the analysis.

Author Response

This is a well-written paper using DTI to assess the effect of EPO in preterm infants. The method is well written, the statistical model is strict, and the results are well presented. But I have several suggestions for the paper:

Thank you for your review and taking to the time to help us improve our summary of DTI methodology and findings in the PENUT cohort. We have provided point-by-point responses below as well as highlighted relevant changes in the manuscript.

I really appreciate the researchers consider the site and scanner difference, when designing the study. When saying Siemens and Philips values are similar (line 99), could you provide the mean of two scanners, and p-value of the comparison?

The FA means of the Siemens and Philips scanners were 0.198 and 0.194, respectively, with a two-tailed P value of 0.32 (no significant difference). This information was added to the manuscript (Page 3 Lines 104-106).

Co-registration of infant's brain is a difficult job. Could you provide the registration result to convince us the registration is accurate? Otherwise, the detected regions may be biased.

ANTS software MeasureImageSimilarity metric was used to determine goodness of co-registration where 1.0 is considered a perfect score: Mean metric for 219 subjects = 0.968, standard deviation for metric = 0.0024 (Page 4 Lines 147-149).

Command used to measure metric: /usr/local/ANTs-2.1.0-rc3/bin/MeasureImageSimilarity 3 1 standardimage.nii.gz testimage.nii.gz  >> test.txt

The definition of five ROIs. Please clarify the details about how to get these regions. AAL has lots of regions, why only these regions are included? Is there a threshold of voxel count? And, please also provide the accurate name of these regions. AAL has anterior/mid/posterior cingulate gyrus, as well as lots of occipital regions.

The ROIs in the occipital and cingulate areas were based on significant large clusters (with corrected pvalues < .02) found on the axial diffusivity TFCE randomise statistical map created by comparing the 2 main groups. These ROIs/clusters were near AAL regions described as follows: AAL region 50 superior occipital gyrus, AAL region 52 middle occipital gyrus, AAL region 34 middle cingulate cortex, AAL region 20 supplemental motor area. The ROIs chosen in the basal ganglia were based on prior reports of brain injury found in the basal ganglia (please see reference added below), and these regions were close to AAL region 73 Putamen Left and AAL region 74 Putamen Right (Page 4, Lines 167-174).

Volpe, J.J., Brain injury in premature infants: a complex amalgam of destructive and developmental disturbances. Lancet Neurol, 2009. 8(1): p. 110-24.

Clustering coefficient is just one parameter to represent information transfer ability within a graph. There are other parameters, such as local/global efficiency, small worldness. These are also widely used in previous papers. Why only considered clustering coefficient in this study?  Specifically, clustering coefficient is associated with local efficiency. In other word, only using clustering coefficient to represent information transfer efficiency is relatively biased.

We agree that clustering coefficient is just one of the structural connectivity parameters available for analysis. As an indicator of network segregation, clustering coefficient measures functionally distinct networks that have been linked to separate and measurable cognitive processes (Watts 1998, Smyser 2019). We chose to focus specifically on this parameter as changes to these cognitive processes have been demonstrated in the pediatric literature, with strong links to measurable neurodevelopmental and behavioral outcomes including internalizing and externalizing behaviors as measured by the Child Behavior Checklist at 2 and 4 years of age (Wee 2017), as well as reading dysfunction in school-age children (Richards 2018).

Preliminary investigations evaluating the structural connectome of healthy or mildly preterm infants have begun to shed light on the natural development of these structural networks (Tymofiyeva 2013, Batalle 2017). The effect of extremely preterm delivery on this process is unknown, and normative values in this patient population at term-equivalent age are missing. Thus, we hoped to add to this growing field.

The above explanations were added to the manuscript (Page 4 Lines 181-192).

We acknowledge that clustering coefficient is associated with inter-network functionality and have thus edited our verbiage to more carefully describe it (Page 2 Lines 55-57).

Watts, D.J. and S.H. Strogatz, Collective dynamics of 'small-world' networks. Nature, 1998. 393(6684): p. 440-2.

Smyser, C.D., et al., Neonatal brain injury and aberrant connectivity. Neuroimage, 2019. 185: p. 609-623.

Wee, C.Y., et al., Neonatal neural networks predict children behavioral profiles later in life. Hum Brain Mapp, 2017. 38(3): p. 1362-1373.

Richards, T.L., et al., Brain's functional network clustering coefficient changes in response to instruction (RTI) in students with and without reading disabilities: Multi-leveled reading brain's RTI. Cogent Psychol, 2018.  

Tymofiyeva, O., et al., A DTI-based template-free cortical connectome study of brain maturation. PLoS One, 2013. 8(5): p. e63310.

Batalle, D., et al., Early development of structural networks and the impact of prematurity on brain connectivity. Neuroimage, 2017. 149: p. 379-392

Setting FC matrix at 10% sparsity is arbitrary. A strict way is to test the robustness of the findings at different sparsities (i.e. 5%, 10%,15%,20%,25%,30%). Or use the AUC (1%-50% sparsity), instead of clustering coefficient from a specific sparsity, in the analysis.

Our choice of a 10% threshold was intentional and based on a lovely analysis by Wang et al supporting that use of thresholds between 10% and 50% for graph analyses in children (Wang 2017). Additionally, use of this threshold has been used in prior pediatric literature (Richards 2018) (Page 5 Lines 202-203). That said, we acknowledge that authors can choose to report results for a range of sparsities and indeed this may have made our findings stronger. We appreciate the feedback and will incorporate this perspective going forward.

Wang, J., Q. Dong, and H. Niu, The minimum resting-state fNIRS imaging duration for accurate and stable mapping of brain connectivity network in children. Sci Rep, 2017. 7(1): p. 6461.

Richards, T.L., et al., Brain's functional network clustering coefficient changes in response to instruction (RTI) in students with and without reading disabilities: Multi-leveled reading brain's RTI. Cogent Psychol, 2018. 5.

Round 2

Reviewer 2 Report

All answers are acceptable.

There is one incomplete sentence at Page 4, Line 212 (...10%s threshold has been previously published and [45, 51]. Brain...)

Author Response

The sentence has been edited to read: "The clustering coefficients of the regions were based on the full-brain connectivity network regions thresholded at 10% sparsity – network efficiency studies support thresholds from 10% to 50%, and a 10% threshold has been previously utilized for graph analyses in children [45, 51]."

Reviewer 3 Report

Minor: Please revisit page 5 lines 210-12 to correct an editorial error. 

Author Response

(The authors gave the same response as above.)

Reviewer 4 Report

Thanks a lot for the feedback. This manuscript has been revised significantly. I don't have further questions about this paper. 

Author Response

Thank you for your positive opinion.